# Slow Evolution toward “Super-Aggregation” of the Oligomers Formed through the Swapping of RNase A N-Termini: A Wish for Amyloidosis?

**DOI:** 10.3390/ijms231911192

**Published:** 2022-09-23

**Authors:** Giovanni Gotte, Elena Butturini, Ilaria Bettin, Irene Noro, Alexander Mahmoud Helmy, Andrea Fagagnini, Barbara Cisterna, Manuela Malatesta

**Affiliations:** 1Department of Neuroscience, Biomedicine and Movement Sciences, Biological Chemistry Section, University of Verona, Strada Le Grazie 8, 37134 Verona, Italy; 2Department of Neuroscience, Biomedicine and Movement Sciences, Anatomy and Histology Section, University of Verona, Strada Le Grazie 8, 37134 Verona, Italy

**Keywords:** ribonuclease A, protein oligomers, 3D domain swapping, protein aggregation, amyloidosis

## Abstract

Natively monomeric RNase A can oligomerize upon lyophilization from 40% acetic acid solutions or when it is heated at high concentrations in various solvents. In this way, it produces many dimeric or oligomeric conformers through the three-dimensional domain swapping (3D-DS) mechanism involving both RNase A N- or/and C-termini. Here, we found many of these oligomers evolving toward not negligible amounts of large derivatives after being stored for up to 15 months at 4 °C in phosphate buffer. We call these species super-aggregates (SAs). Notably, SAs do not originate from native RNase A monomer or from oligomers characterized by the exclusive presence of the C-terminus swapping of the enzyme subunits as well. Instead, the swapping of at least two subunits’ N-termini is mandatory to produce them. Through immunoblotting, SAs are confirmed to derive from RNase A even if they retain only low ribonucleolytic activity. Then, their interaction registered with Thioflavin-T (ThT), in addition to TEM analyses, indicate SAs are large and circular but not “amyloid-like” derivatives. This confirms that RNase A acts as an “auto-chaperone”, although it displays many amyloid-prone short segments, including the 16–22 loop included in its N-terminus. Therefore, we hypothesize the opening of RNase A N-terminus, and hence its oligomerization through 3D-DS, may represent a preliminary step favoring massive RNase A aggregation. Interestingly, this process is slow and requires low temperatures to limit the concomitant oligomers’ dissociation to the native monomer. These data and the hypothesis proposed are discussed in the light of protein aggregation in general, and of possible future applications to contrast amyloidosis.

## 1. Introduction

The events governing protein self-association can favor the formation of amyloid fibrils and are among the most studied life science topics of the last 20–25 years [1,2,3]. In particular, the three-dimensional domain swapping (3D-DS) mechanism associated with protein oligomerization, in some cases related to massive supramolecular aggregation [4,5,6].

In this context, monomeric (M) bovine pancreatic ribonuclease A (RNase A, EC 3.1.27.5), the proto-type of the secretory pancreatic-type (pt)-RNase super-family [7], was known to self-associate upon interaction with substrate(s) [8]. Moreover, RNase A forms different conformers of N-termini- and/or C-termini-swapped dimers, trimers, tetramers, and larger oligomers [9,10,11] after lyophilization from 40% acetic acid (HAc) solutions and subsequent solubilization in sodium phosphate (NaPi) buffers, according to the protocol described in the 1960s by S. Moore and colleagues [12]. The same domain-swapped oligomers can be produced by heating highly concentrated solutions of the protein in various solvents, such as 40% ethanol (EtOH) [13,14].

The crystal structures of the dimers formed through the swapping of their N-termini (residues 1–15) or C-termini (residues 116–124), and respectively called N-dimer, or N_D_ [15], and C-dimer, or C_D_ [16], have been solved. Moreover, the structure of a cyclic exclusively C-swapped trimer (C-trimer, or C_T_) is available [17]. Then, 3D models have been proposed for a RNase A N+C-swapped trimer [17,18], and for many N+C-swapped tetramers, or larger multimers on the basis of experimental data [10,11,17,18,19]. Overall, at least two or more linear or/and cyclic oligomeric conformers of the enzyme can be purified either by size-exclusion chromatography (SEC, Figure 1A) or by cation-exchange chromatography (Figure 1B) [20]. Indeed, the polarity of both RNase A N- and C-termini [21], and glycosylation and/or deamidation involving key-Asn/Gln residues [22,23], can affect its self-association through 3D-DS.

Notably, RNase A displays many amyloidogenic segments [24], including the S-peptide that corresponds to its first 20 residues, and spontaneously undergoes fibrillation when it is cleaved from the rest of the protein [15]. Nevertheless, amyloidosis involving entire wt RNase A has never been registered [24] until 2021 [25]. Nevertheless, RNase A variants displaying poly-G or poly-Q insertions, respectively, located at the edges or inside the loops connecting the protein core to the N- and C-termini can spontaneously produce cross-β amyloid-like fibrils [26,27]. However, Noji and co-workers recently found that also wt-RNase A undergoes amyloidosis but only upon overcoming the super-saturation barrier [25].

Importantly, the oligomeric or pre-fibrillary supramolecular adducts, and not the mature fibrils, are considered the actual toxic species inducing neurodegenerative diseases [28], although this aspect is still debated [29]. Moreover, in amyloidosis contexts, the term “oligomer” is often assigned to very large (proto- or pre-fibrillar) species, and not only to dimers, trimers, or small multimers. Indeed, for the term “large or very large multimer(s)”, only oligomers or multimers might be more appropriate for these big aggregates, although the molecular weight (MW) limits distinguishing oligomers from large or very large multimers, or even pre-fibrillar species, are certainly difficult to be settled [30,31]. Despite this, a valuable classification for these terminologies was proposed by M. Fändrich in 2012 [32].

In this scenario, we casually found that some of the mentioned RNase A domain-swapped oligomers produced not negligible amounts of very high MW species after being stored for months at 4 °C. Therefore, we deepened the analysis of the long-term behavior of these small oligomers, i.e., from dimers to hexamers, purified from one another and from the residual monomer. To do so, we stored them at 4 °C, and at a concentration between about 0.05 and 0.10 mg/mL in 0.10–0.12 M sodium phosphate (NaPi) buffer. After about 6 to 15 months, some of the oligomers formed aggregate derivatives characterized by very high MWs. Hence, we call these species “super-aggregates” (SAs). Notably, SAs originated only from RNase A oligomers characterized by the swapping of their N-termini, or of their N+C-termini as well, while neither from the native monomer nor from C_D_, C_T_, i.e., species exclusively formed by swapping of their C-termini. Even if we did not detect the presence of amyloid-like [33] fibrils, we discuss here the implication deriving from the detection of SAs in light of the phenomena characterizing the context of massive protein aggregation in general.

**Figure 1 ijms-23-11192-f001:**
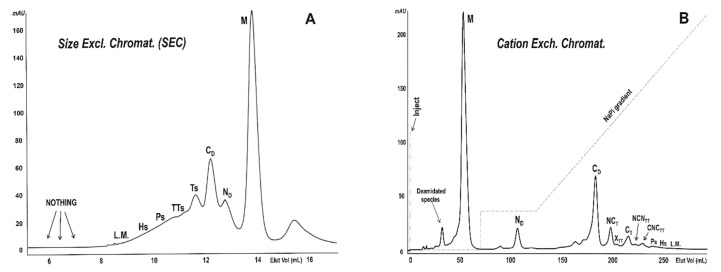
Chromatographic purification of the RNase A domain-swapped oligomers [9,20]. RNase A oligomerization was induced by lyophilizing the enzyme from 40% HAc solutions and re-dissolving it in 50 mM NaPi buffer, pH 6.7 [12]. Qualitatively similar results (not shown) were registered by incubating highly concentrated 40% EtOH solutions of the enzyme at 60 °C for up to 2 h [13,14]. (**A**) Size-exclusion chromatography (SEC): 1.5 mg of the RNase A mixture was injected onto a Superdex 75 HR 10/300 column and eluted at a 0.20 mL/min flow rate. No species were eluted around 6–7 mL. (**B**) Cation-exchange chromatography: 12 mg of the same mixture was injected onto a Source 15 S 16/10 column equilibrated with 70 mM NaPi, pH 6.7. After eluting the monomer, the NaPi concentration was increased to 90 mM to elute the N-dimer. Then, a 90–180 mM NaPi linear gradient (dashed line) was applied to elute all other RNase A oligomers and make possible a satisfactory separation of the N-swapped oligomers from the C-swapped or N+C-swapped ones [9,20]. Legend: M, RNase A monomer; N_D_, N-swapped dimer; C_D_, C-swapped dimer. The two dimers can be distinguished with SEC following that previously indicated in [20] and, above all, in [34]; Ps and Hs domain-swapped pentamers and hexamers, respectively; L.M., large multimers traces. In panel A: Ts and TTs, domain-swapped trimers and tetramers, respectively. In panel B: NC_T_, N+C-swapped trimer; C_T_, C-swapped trimer; X_TT_, NCN_TT_ and CNC_TT_, N+C-swapped linear tetramers. The qualitatively similar profiles obtained upon heating RNase A with 40% EtOH [13,14] are not reported.

## 2. Results

### 2.1. Production and Purification of RNase A Oligomers

Upon 40% HAc lyophilization and subsequent solubilization in aqueous NaPi buffer [12,35], RNase A undergoes controlled self-association and produces 3D domain-swapped oligomers up to traces of tetradecamers detected through mass spectrometry [9,10,36]. The same oligomers are formed by heating the enzyme up to 2 h at 60–70 °C in various solvents, such as 40% EtOH, but only if the solution is highly concentrated [13,14]. In both cases, RNase A oligomers can be dissociated by heat [15] while they are slightly more resistant against urea [14].

We produced the RNase A oligomers by following the protocols mentioned above [12,13]. Then, we separated and purified these species through SEC or, more efficiently, through cation exchange chromatography [20], as shown in Figure 1A,B, respectively.

Panel A shows the formation of RNase A oligomers/multimers but no relevant amounts of species characterized by MW species larger than about 100 kDa (~heptamers) [10,36] nor any species eluted at the dead volume of the Superdex 75 column. Panel B reports the possibility of separating the variety of oligomers that formed through the swapping of the two RNase A N- and/or C-termini with cation-exchange chromatography [20]. In particular, they are dimers (N_D_ and C_D_), trimers (NC_T_ and C_T_), tetramers (X_TT_, NCN_TT_ and CNC_TT_), pentamers (Ps), hexamers (Hs), and traces of larger multimers (L.M.) were separated through cation-exchange chromatography following the protocol illustrated in previous reports [10,11,20].

### 2.2. RNase A Oligomers Characterized by the Swapping of at Least One N-Termini’s Couple Produce Very High MW Species upon 4 °C Long-Term Storage

Each RNase A oligomeric conformer was accumulated and separately collected from one another and from the residual monomer, M. Then, each species was incubated at 4 °C in NaPi pH 6.7, as reported in the Section 4. After 6 to 15 months, each sample was separately concentrated with Amicon 3 (3 kDa c.o.) ultrafilters. We found that turbidity was almost totally eliminated with centrifugation (Appendix A), and the pellet was stored for further analyses.

Each concentrated oligomer was re-chromatographed with the two Superdex-75 or 200 HR 10/300 SEC columns, whose calibration curves are reported in Appendix A. As shown in Figure 2, panels A-H, after repeated tests, only the oligomers containing at least one swap of their N-termini formed detectable amounts of very large species eluting around the dead volume of both SEC columns. We call these super-aggregates (SAs).

Considering the results obtained with all columns, we estimated their MW to comprise, SA-1 to SA-3, between about 4.2 × 10^6^ and 2 × 10^5^ Da. The Superdex 75 column, better than the Superdex 200, allowed us to partially resolve two, sometimes three, different parts of SAs peaks, which we call SA-1,2,3. Considering the elution volumes visible in panels A, C, E–G of Figure 2, an MW value of about 2.5 × 10^6^ could be estimated from the elution volume of the SA-2 apex, corresponding to aggregates that could apparently contain about 180 RNase A units. Notably, only the species displaying the swapping of their N-termini, in particular N_D_ [15], NC_T_ [17,18], N+C-(swapped)-tetramers (TTs) [11,18,19], or N+C-pentamers (Ps) or N+C-hexamers (Hs) [10,11], produced SAs derivatives (Figure 2A,C,E–G, respectively). Conversely, neither the monomer (M) nor both the C_D_ [16] or C-swapped-only cyclic trimer, C_T_ [17], produced SA species (Figure 2B,D,H).

Notably, in all panels of Figure 2, except in panel H, relative to the monomer, species lighter than the original oligomer, which is marked in bold, are visible. The presence of these species is ascribable to partial dissociation of the original oligomer [10,18,37], dissociation that in turn is concomitant with SAs formation. Indeed, it is known that RNase A oligomers are metastable but, when they are not subjected to drastic conditions, they dissociate slowly and progressively pass through smaller oligomers before coming back to M [10,18].

We then tested the total mix of oligomers, except the dimers (see Figure 1), stored again at 4 °C for 12 months, with the “Increase” version of both Superdex 75 and 200 columns. Many species eluted again at their dead volume, and we found (Figure 2I,J, black curves) qualitatively the same results obtained with each N-swapping-containing oligomer analyzed alone (panels A, C, E–G). In this case, we collected one single SA peak, although showing a visible fronting (formerly, SA-1) with the Superdex 75 Increase column, while we divided two fractions (SA-1 and SA-2) with the Superdex 200 one. Hence, in this case, SAs were slightly better resolved by the latter column than by the former. We underline here that even the two most modern SEC columns did not allow us to completely separate the SAs subpopulations, but this possibility is beyond the scope of our study. This argument is enforced by a huge amount of literature data in which large protein aggregates or amyloid fibrils are described by exploiting their electron-microscopy dimensions or assemblies [32,38] or after considering only a range of subunits forming them rather than their precise number, and, therefore, their precise MW [31].

**Figure 2 ijms-23-11192-f002:**
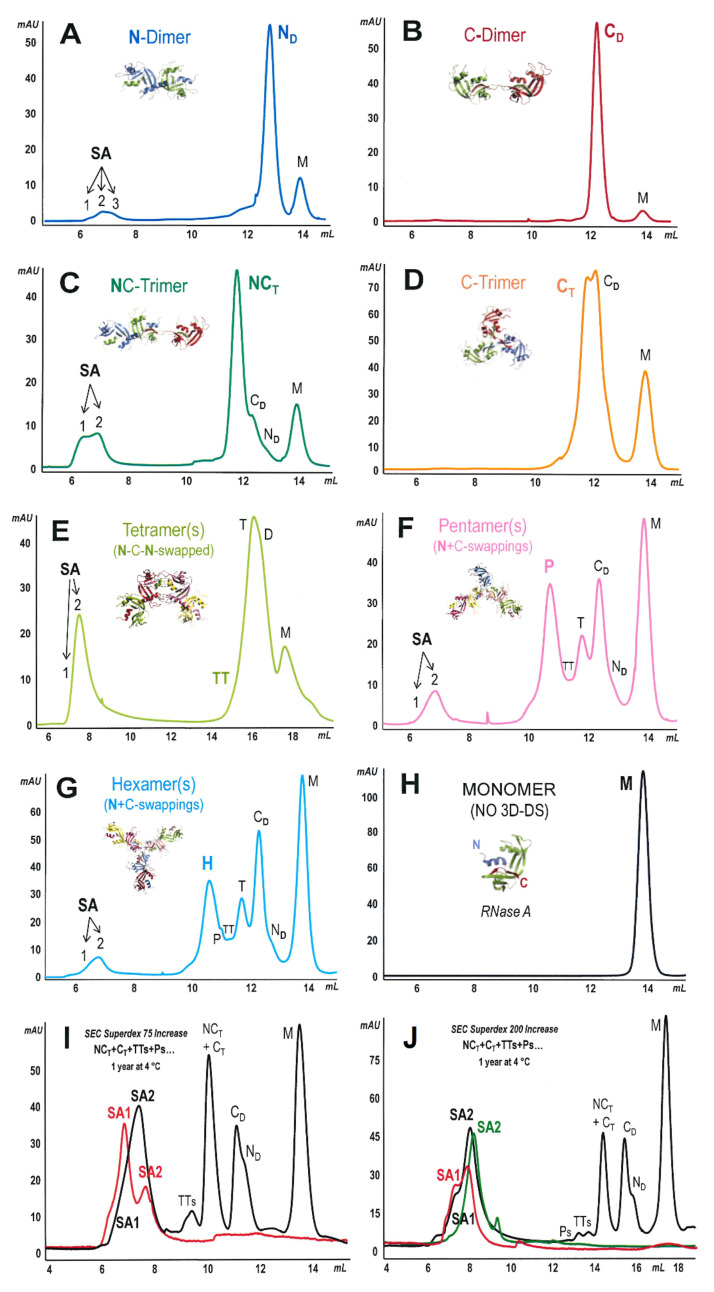
SEC purification and analysis of RNase A super-aggregates (SAs). Each RNase A oligomer, indicated in bold in each panel, and purified as in Figure 1B, was stored for 12 months in NaPi at 4 °C and was concentrated and injected onto a Superdex 75 HR 10/300 (**A**–**D**,**F**–**H**), Superdex 200 HR 10/300 (panel **E**) SEC columns, both equilibrated with 0.2 M NaPi buffer, pH 6.7. All species, eluted at 0.2 mL/min, are labeled as in Figure 1. The relative structures reported in the panels were built with PyMol: (**A**) N-dimer, PDB 1A2W [15]; (**B**) C-dimer, PDB 1F0V [16]; (**C**) NC-trimer, model built in [17], in line with the experimental data reported in [18]; (**D**), C-trimer, PDB 1JS0; (**E**) N-C-N-tetramer, model built in [11], in line with the data reported in [19,39]; (**F**,**G**) swapped models built in [11], representative of the numerous N+C-swapped pentameric and hexameric conformers characterized in [10]; (**H**) RNase A (M), PDB 7RSA. (**I**,**J**) The entire mixture of RNase A oligomers ranging from N+C-swapped trimers to hexamers (monomer and dimers excluded) were chromatographed, after 12 months of storage at 4 °C, with the “Increase” versions of the Superdex 75 and 200 SEC columns, respectively (black curves): only SAs were collected, concentrated with Amicon 100 (c.o. 100 kDa), and chromatographed again in the same two “Increase” columns (red and green curves). In all panels, except (**B**,**D**,**H**), super-aggregates (SA-1,2,3) are eluted around 6-8 mL, corresponding to the dead volume of the columns. In each panel, except (**E**,**I**,**J**), the pattern is representative of at least three independent experiments.

Finally, after discarding the oligomers and the monomer, using Amicon 100 (c.o. 100 kDa) ultrafilters, we concentrated the SAs eluted from both columns, and we chromatographed them again using the same columns. The results visible in the red curves of both panels, and in the green pattern of panel J (SA-2) show no significant quantitative and qualitative differences concerning SAs as compared with the black curves. The red curve of panel I shows in this case a split, doubled peak; however, the elution volumes of both fractions fall around one of the single, enlarged but unresolved peak of the black curve. In panel J, the peak of the red curve (SA-1) is split again, in this case because it was polluted by the more abundant SA-2 in the first separation (black curve). Again, both SA-1 and SA-2 (red and green curves) eluted around their elution volumes relative to the first separation of the black curve. Overall, these results indicate that all SAs were retained by the 100 kDa ultrafilters, and confirm they are very large aggregates. Incidentally, considering that the “200” column can partially separate species characterized by MW reaching 600 kDa (manufacturer declaration), we envisage the MW of a relevant part of SAs might overpass this value.

### 2.3. The UV-Vis Spectra of SAs Are Altered with Respect to RNase A Monomer and Oligomers

The UV-vis profiles of SA-1,2,3 were evaluated after eliminating turbidity as much as possible using centrifugation (Appendix A) while saving the pellet for additional analyses. SAs showed an altered spectrophotometric profile (Appendix A) with respect to RNase A M and N_D_ and C_D_ (Appendix A). In particular, the A_260_/A_280_ ratio of all SAs reported in Table 1 is higher than the ones relative to the two dimers. The increase in the A_260_/A_280_ ratio is ascribable to the presence of large aggregates. Indeed, aggregation increases the light scattering with a decrease in the wavelength, according to the Rayleigh law. Finally, and notably, N_D_ always shows a higher A_260_/A_280_ ratio than C_D_ (Appendix A, Table 1).

**Table 1 ijms-23-11192-t001:** Biophysical parameters and enzymatic activity of RNase A species.

RNase A Species	M.W. (Da)	SEC Elut. Vol. ± SD (mL) *	“Kunitz” Activity **	A_260_/A_280_
Monomer (M)	13,686	13.80 ± 0.04	100	0.414/0.795 = 0.52
RNase S	13,686	13.68 ± 0.09	See [35]	N.D. ***
S-protein	11,540	14.11 ± 0.05	N.D. ***	N.D. ***
N-dimer (N_D_)	27,372 ****	12.56 ± 0.07	See [9,14]	0.332/0.425 = 0.78
C-dimer (C_D_)	27,372 ****	12.21 ± 0.08	See [9,14]	0.256/0.451 = 0.57
NC-trimer (NC_T_)	41,058 ****	11.78 ± 0.05	See [9,20]	N.D. ***
C-trimer (C_T_)	41,058 ****	11.78 ± 0.05	See [9,20]	N.D. ***
SA-1	N.D. ***	6.22 ± 0.12	6.4 ± 0.9	0.147/0.116 = 1.27
SA-2	N.D. ***	6.90 ± 0.08	9.5 ± 0.6	0.187/0.145 = 1.29
SA-3	N.D. ***	7.21 ± 0.10	12.4 ± 1.3	0.123/0.102 = 1.21

* Elution volumes measured with the Superdex 75 HR 10/300 analytical column. ** Enzymatic activity spectrophotometrically measured at 300 nm with yeast RNA as substrate following the protocol described by Kunitz [40]. *** N.D.: Not Determined. **** The two dimers and the two trimers, although displaying the same MW, can be separated and distinguished by SEC or especially by cation-exchange chromatography [9,20,34]. The legend code relative to RNase A species is the same as in Figure 1 and Figure 2.

### 2.4. Do SAs Actually Derive from Oligomers of Unaffected RNase A?

We deepened the analysis of SAs considering the two following limitations: (i) the scarce SAs amounts obtained, and (ii) the impossibility, with both SEC columns, to completely purify one SA species from the other two. Therefore, we performed many of the studies only with the most abundant species, i.e., SA-2, obviously foreseeing a probable contamination of the two other SAs.

First, we performed a test to ensure SAs do not derive from the RNase A S-peptide derivative [15]. To do so, we independently analyzed the S-peptide/S-protein RNase A derived species with SEC, and compared them with RNase A native monomer in order to exclude the occurrence of RNase A degradation. Then, we qualitatively measured the SAs enzymatic activity, analyzed the effect of heat on SA-1 and SA-2, and immunoblotted SA-2 with the RNase A antibody, either under denaturing or non-denaturing conditions.

### 2.5. S-Peptide/S-Protein Analyses Exclude That SAs Derive from RNase A S-Peptide

Traces of subtilisin protease are known to catalyze the cleavage of the RNase A Ala20-Ser21 peptide bond, or, to a lesser extent, the Ser21-Ser22 one [41]. Consequently, a 20/21 residues peptide, representing the RNase A N-terminal end (S-peptide), accompanied with a 21/22-124 residues S-protein, is produced [41]. The two fragments form the so-called RNase S derivative through electrostatic association but can alternatively be separated through EtOH precipitation [41]. Notably, the S-peptide contains the _15_SSTSAA_20_ sequence [42], which can spontaneously undergo amyloid-like [33] fibrillation [15,42]. Considering that subtilisin more efficaciously cleaves the RNase A N-swapped oligomers than the C-swapped ones [18,43], we aimed to ensure its traces were absent in our samples, and other potential degrading pollutants. Otherwise, a concomitant RNase A degradation would have produced S-protein plus S-peptide derivatives, in this way triggering the formation of SAs, if not fibrils [15].

Therefore, we isolated the S-protein through EtOH-precipitation of the S-peptide [41], and we compared its chromatographic behavior with that of native RNase A (M). The elution volume of M recovered from the Superdex 75 SEC column after long-term 4 °C storage was different from the RNase S one, and, above all, from that of S-protein (Figure 3A). Then, Figure 3B shows that the mix of RNase A plus S-protein could be separated, although partially, presenting a split monomer peak.

Instead, in the case of SAs formed by RNase A oligomers, no splitting of the M peak relative to the partial dissociation of the latter ones is detectable in all panels of Figure 2. Moreover, the elution volume of M is always different from that of RNase S visible in Figure 3. Again, if the residual M recovered from oligomers’ dissociation in Figure 2 was actually the S-protein, which is devoid of the RNase A N-terminus, no SAs would have been formed since it is known that RNase S and S-protein can produce, under the conditions used for RNase A, only a C-swapped dimer, and not larger oligomers [35].

Therefore, and overall, we can state, although indirectly, that RNase A oligomers were not degraded here to S-protein + S-peptide by any possible pollutant upon their 4 °C long-term storage. Consequently, we can deduce that SAs do not derive from S-peptide [15] nor from S-protein but, instead, from oligomers of the entire, unaffected RNase A, i.e., containing N-swapped subunits.

### 2.6. SAs Retain Low Enzymatic Activity

We measured the enzymatic activity of SA-1,2,3, although it was reciprocally partially contaminated. We used yeast RNA, the standard substrate for RNase A, as indicated by Kunitz [40]. We found that, considered equal to 100 RNase A (M) activity, SAs species retain activity estimated to range only from about 6 to 13% of the native protein, as reported in Table 1.

Even if these data must be considered as only estimations, due to the difficulty in accurately measuring the SAs amount injected in the assays (see Section 2.3), it confirms that SAs are RNase A derivatives. Instead, they do not derive from RNase A degradation, which would have in turn induced the detachment of the peptide containing the His12 catalytic residue. Instead, we can ascribe these results to the possibility of only few active sites being accessible for the substrate, whereas many of them might be hindered by the large structure of the aggregate(s).

### 2.7. SAs Are Partially Resistant to Heat while Immunoblotting Confirms They Are RNase A Derivatives

We again chromatographed SA-1 and -2, this time mixed together and stored at 4 °C for 1 month (Figure 4A), or alternatively after boiling them (Figure 4B,C, respectively). Some unresolved smaller oligomeric residues were visible only after boiling while only a small amount of M was detected. Then, SA-2 was analyzed, together with RNase A M, N_D_, and C_D_, with cathodic PAGE, i.e., under non denaturing conditions [34,44], or with SDS-PAGE (Figure 4D,E).

In both cases, we immunoblotted the species with an RNase A-active antibody. The first three lanes of Figure 4D show, as expected, the antibody-reactive bands of M, C_D_, and N_D_ [18,20,34]; however, in the last lane, antibody-reactive species are also visible for SA-2, although with low intensity. The faint staining could be ascribed to the epitope somehow being sterically hindered in the large SA aggregates. Instead, the immunoblotting performed under denaturing SDS-PAGE, i.e., reducing and boiling conditions (Figure 4E), showed an intense band in the SA-2 lane, indicating the presence of a large amount of RNase A monomeric derivative. The same occurs for the dissociating dimers, and, obviously, for M. By contrast, only traces of oligomers resisting dissociation are visible in the SA-2 lane. We can therefore envisage that the denaturant-reducing SDS plus β-mercapto-ethanol mixture increase SAs dissociation while boiling alone is not sufficient to significantly make SAs dissociate to M (Figure 4B,C).

**Figure 4 ijms-23-11192-f004:**
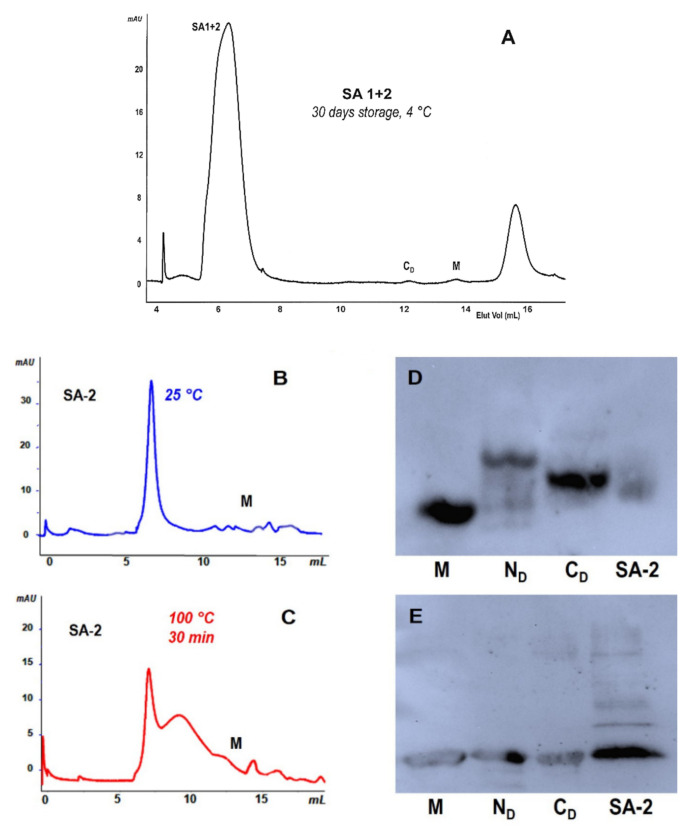
Stability and immunoblot analysis of SA-2. (**A**) SEC analysis of SA-1 + SA-2 after 30-day storage at 4 °C in 0.2 M NaPi, pH 6.7. (**B**,**C**) SEC analysis of SA-2 after 2-h storage at 25 °C or after 20-min incubation at 100 °C, respectively. (**D**,**E**) Western blots of SA-2 performed in parallel with RNase A monomer (M), N-dimer (N_D_), and C-dimer (C_D_) after cathodic PAGE was performed, respectively, under non-denaturing conditions, in which C_D_ displays an electrophoretic mobility higher than N_D_ [34], or after SDS-PAGE was performed under reducing conditions, in which all oligomers dissociate to the monomer.

### 2.8. Thioflavin-T (ThT) Fluorescence Emission Indicates That SAs Are Large Aggregates Not Fibrils

In order to unveil whether SAs are characterized by the presence of amyloid-like fibril structure(s), we performed fluorescence measurements with ThT and testing in parallel RNase A multimers (hexamers + pentamers + tetramers), trimers, dimers, and monomer. To do so, we concentrated the mentioned species with Amicon 3 ultrafilters (3 kDa c.o.) as indicated in the Section 4. The possibility of detecting wt-RNase A fibrils in some way parallels the results recently presented by Noji et al. [25], and this is somehow in contrast with the statement of Eisenberg and colleagues, who reported that wt-RNase A is a “self-chaperone”, since its structure hinders a possible fibrillogenic destiny [26]. Indeed, this inability disappears when many short peptides actually undergo fibrillation when they are detached from the entire RNase A “jail” structure [24,42,45].

In our case, the ThT fluorescence intensity emission induced by SAs or by other RNase A supramolecular derivatives obtained either upon acid lyophilization [12] or after heating the enzyme in 40% EtOH solutions [13] was not negligible. The results relative to both conditions visible in Figure 5A,B show the fluorescence intensity at 482 nm increases, almost regularly, from RNase A monomer, for which it is close to zero, to the pentameric or hexameric conformers [10].

A decrease was instead registered, with SAs forming either under HAc-lyophilization or EtOH-heating. However, the intensity values were higher than the ones provided either by dimers or trimers, and this consideration must also take into account that the concentration of SAs is difficult to be precisely settled (see Appendix A). We also tested the possible interaction with ThT of the insoluble residues recovered from SAs and partly re-dissolved in 10 mM NaPi, pH 6.7. Panel A shows the residue deriving from acid lyophilization is ThT-positive while the one deriving from EtOH thermal incubations induces a lower fluorescence intensity, which is in turn higher than the corresponding RNase A trimers, dimers, and M. Since, in both cases, N_D_ is more ThT-active than C_D_, we underline that N_D_ is characterized by a six consecutive β-strand extension [15] while in C_D_, only three consecutive β-strands are present [16], such as in M. Hence, the actual presence in N_D_ of a minimal four-strand extended β-sheet, which represents the requisite for a ThT-productive binding [46], justifies this result.

Then, we found that the ThT fluorescence intensity decreases from large oligomers/multimers to the, in turn, larger SAs, suggesting these derivatives are not annular, proto-fibrillar species [47]. However, they probably derive from multimers characterized by ThT fluorescence emission levels close to the ones characterizing large pre-fibrillary multimers [47]. Considering that annular pre-fibrillary aggregates should provide ThT fluorescence intensity values similar to amyloid-like fibrils [32], we hypothesize that SAs are large, although neither fibrillary nor annular RNase A, aggregates. Instead, they could be similar to the so-called “amorphous aggregates” [30,31,32], the formation of which we found to be promoted by the 3D-DS of the N-termini of the enzyme.

Interestingly, as it was recently detected for the dimers, the two protocols inducing RNase A oligomerization [12,13] did not significantly affect the ThT results obtained with all species, except for the insoluble residues [14].

### 2.9. Transmission Electron Microscopy (TEM) Indicates SAs Are Circular, Very Large Aggregates

In order to gain additional structural information, we analyzed SA-2 with TEM together with RNase A multimers (tetramers+pentamers+hexamers) and with the insoluble residues. Figure 6 reports that RNase A multimers (panel A), SA-2 (panel B), and insoluble residues (panel C) form circular aggregates. This occurs after RNase A self-association upon 40% HAc lyophilization or thermal incubation in 40% EtOH [12,13]. The average diameters of the mentioned species were measured and their size distributions are reported in Figure 6D. In particular, the diameter(s) of the multimers ranged from 4.4 to 6.4 nm while SAs ranged from 15.7 to 21.0 nm and insoluble residues from 10.7 to 49.7 nm. The size of the species belonging to different samples was statistically different from each other (one-way ANOVA: *p* < 0.001; Bonferroni post-hoc test: *p* < 0.01 for all comparisons). The diameter of these aggregates is generally not higher than 50 nm, as it occurred for the species considered by M. Fändrich to be “oligomers” or, better, “non-fibrillar aggregates” [32].

Overall, and finally, these results suggest again that the large aggregates studied here, although displaying almost circular shapes, should be different from the “annular protofibrils” or, again, from the “amyloid oligomers” described by Kayed et al. [47].

### 2.10. Kinetics Evaluation of RNase A SAs’ Formation

In order to better understand why the low-concentration RNase A N-swapped oligomers can form large aggregates after 4 °C long-term storage, we analyzed the chromatographic behavior of the N+C-N-swapped plus C+N+C-swapped RNase A tetramers mixture [9,11,18] after 2 and then after 15 months (Figure 7A, red and blue curves, respectively). The results obtained suggest that oligomers’ super-association results in SAs formation but concomitantly with dissociation or regression toward M (see Figure 2). This result is in line with previous evidence, but in other cases, the oligomers’/multimers’ dissociation to M was eased by increasing temperatures [15,18,21]. In the way, this last evidence impedes us from speeding up the evolution of the oligomers toward SAs by increasing the temperature of storage because, in this way, the oligomers’ dissociation [15] prevails on SAs formation.

**Figure 7 ijms-23-11192-f007:**
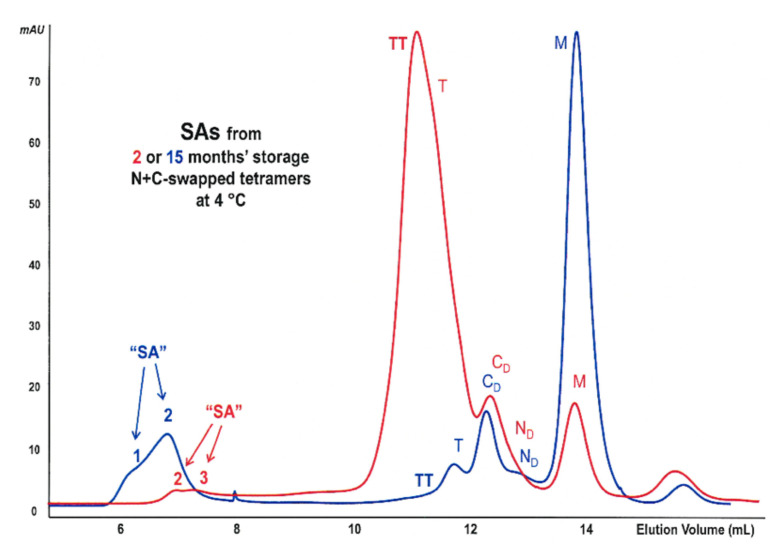
Kinetic and structural analysis of SAs. SEC analysis to measure the amount of SAs formed from N+C-N- plus C-N-C-swapped RNase A tetramers [9,11,18] after 2- or 15 month-storage at 4 °C (red and blue curves, respectively).

## 3. Discussion

The present study focused on the possible long-term evolution of RNase A oligomerization. Protein self-association can sometimes drive protein fibrillation, either in vitro or in vivo [33], with the latter affecting patients with systemic or localized amyloidoses [48,49]. We report here that RNase A, which can be considered as a model for controlled protein self-association occurring through 3D-DS [11,19,50], shows a somewhat strange behavior. Indeed, we found that many RNase A domain-swapped oligomers or multimers (Figure 1) [10,11] can in turn evolve to low amounts of larger supramolecular derivatives (Figure 2), which we call super-aggregates (SAs). Interestingly, the slow formation of SAs occurs concomitantly with the dissociation of the oligomers, which in turn drives them back toward the native monomer, M [10,18,21].

Importantly, we found that SAs are different from amyloid fibrils, although they are partially ThT-active, with a fluorescence intensity that generally increases with the dimensions and, hence, with the MW of the RNase A aggregates (Figure 5). However, TEM analysis shows that SAs are not rod-like, linear structures (Figure 6), like fibrils should be [30]. Again, SAs are also almost certainly different from the annular, pre-fibrillary aggregates characterized by Kayed et al. [47] while they are instead similar to the so-called amorphous protein aggregates [30,31,32]. Precisely, SAs are very large (Figure 1 and Figure 6) and circular oligomers/multimers (Figure 6) [32]. This scenario resembles some of the ones displayed by TDP-43, the 43 kDa DNA-binding protein involved in neurodegenerative diseases and able to undergo either fibrillation or amorphous aggregation as well. Indeed, some TDP-43 aggregates are known to resemble amyloid structures [51] while many others do not [52], with this possibility often depending on the presence of point mutations in the protein [53].

Like other amyloidogenic events, SAs form slowly, somewhat unexpectedly, from not highly concentrated RNase A oligomeric precursors. After excluding the origin of SAs from the two S-protein or S-peptide RNase A fragments (Figure 3), the analyses performed found that they form neither from native RNase A monomer nor from the C-swapped dimer, C_D_ [16], nor again from the cyclic C-swapped-trimer, C_T_ [17]. They instead derive from oligomers characterized by oligomers containing at least two subunits undergoing N-termini swapping, such as the N-dimer (N_D_) [15], the N+C-swapped trimer (NC_T_) [17,18], or the multitude of N+C-swapped tetramers, pentamers, hexamers, or even larger RNase A ··÷N+C+N-C÷·· multimers (Figure 2) [10,11,36]. This cannot be surprising, considering that the 16–22 hinge loop responsible for the RNase A N-terminus swapping is in turn able, when detached from the protein, to undergo fibrillation [15,42] thanks to the amyloidogenic _15_SSTSAA_20_ sequence included [24,42]. The same is not true for the 111–124 C-terminus of the enzyme, where fibrillation is triggered only if a poly-Q extension is inserted within the 112–115 flexible loop connecting the terminus with the protein core [27].

The lack of SAs, and obviously of fibrils, deriving from wt RNase A (M) is not surprising, again, if we consider the enzyme is known to act as a “self-chaperone”, hence contrasting the fibrillary destiny [24,26]. To date, the only possibility for RNase A to undergo fibrillation is to overpass the super-saturation barrier, as recently reported by Noji and co-workers [25]. In our case, the “open” conformation [4] of the 16–22 hinge loop, which represents a crucial portion of the open interface [4] stabilizing all RNase A N-swapped oligomers [3,5], might favor part of the oligomers’ population to undergo massive aggregation but without the formation of fibrils.

Hence, also on the basis of the models proposed by McPartland et al. and foreseeing the structure of protein-combining proto-filaments [3], we propose in Figure 8A a tentative representation of the main structural features that could ease the formation of RNase A SAs. Indeed, the so-called “runaway” [54], or “propagated swapping” [3,55] of RNase A N-termini, favors the formation and stabilization of SAs, but we envisage that this may occur together with newly formed “side-by-side” or stacking interactions [3], as proposed and shown in Figure 8A, similarly to what occurs for SOD-1 aggregation [56]. The swapping of RNase A N-termini, leading to the formation of N_D_, provides the minimal four-β-strand repeat required for ThT-positive interaction (Figure 5) and may help a subsequent massive aggregation onset.

This possibility foresees that runaway-propagated N-term swapping (lines connecting the circles) and side-by-side electrostatic, hydrophobic, H-bond, or mixed interactions (differently colored boxes) may co-exist to favor the network formation driving SAs. This model, although tentative, does not contrast with the experimental data found, and can also explain why a “closed” conformation [4] of the RNase A 16–22 loop, such as in the monomer or in the C-swapped-only oligomers, should hinder the evolution toward larger aggregates. Indeed, the C-terminal β-strand of native RNase A forms several and strongly protected hydrogen bonds with the protein core while, on the contrary, the hydrogen bond network connecting the N-terminal α-helix (residues 3–13) with the core is poorer and weaker [57]. Therefore, the detachment from the enzyme core to reach an “open” conformation that helps massive aggregation is easier for the RNase A N-terminus than for its C-terminus, as we previously ascertained [13,21]. Alternative models foreseeing the presence of C-swapped trimeric and/or dimeric (C_T_, C_D_) units seem to be less probable for SAs. Indeed, these units should act like “seeds” to become a sort of intersection knots through which the SA network should be developed via the propagation of the necessary aforementioned N-swapping. This possibility is visible in the models of Figure 8B,C, which were already proposed by us in 2008 [11], when we envisaged the enlargement of oligomers larger than tetramers toward bigger aggregates. These two similar models are less probable if we consider the low concentration of the stored oligomers and the necessary presence of C-swapped structures but are not totally in contrast with the data of the present work. In fact, N+C-swapped pentamers and hexamers [10] are able to form not negligible amounts of SAs, although they are definitely less abundant than the smaller oligomers (see Figure 1).

In this context, it is interesting to mention that we incidentally performed a single test (not shown) to measure the SAs produced by the N_D_ and NC_T_ produced by the S80R-RNase A mutant [37], finding a slightly higher amount of SAs than the one relative to the wt. Since it is known that the S80R mutation favors the RNase A N-swapping more than in the wt [37], this result again indicates that the swapping of RNase A N-termini paves the way to SA formation.

**Figure 8 ijms-23-11192-f008:**
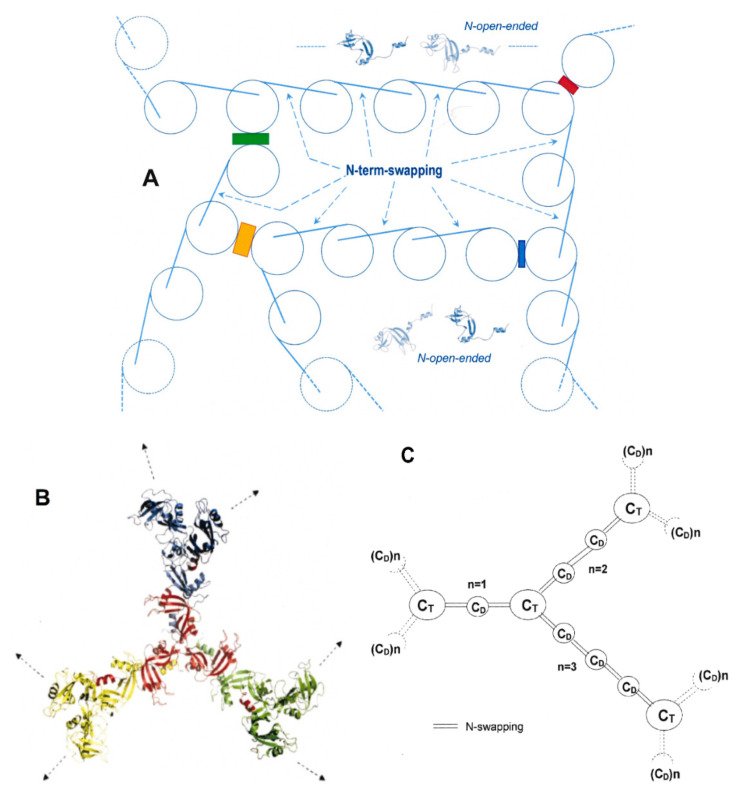
Tentative structural models of SAs suggested by the structural data collected. (**A**) The model foresees the simultaneous presence of 3D-DS indicated by the lines connecting the circles (see the N-term-open structures in light blue, built with PyMol from RNase N_D_, PDB 1A2W [15,17]) representing the subunits, and the “side-by-side” stabilizing interaction contacts [3] indicated by colored boxes. The different colors indicate different interactions, such as H-bond, hydrophobic and/or electrostatic interactions, or a mix of all of them, which might provide the network bridging the unswapped subunits. (**B**,**C**) An alternative model proposed in [11] for large aggregates, starting from C_T_ seeds (colored trimers) and connected by N-swapping (N_D_-like), which in (**C**) can also be propagated through C_D_ units [11]. These two models are possible only if SAs form from oligomers larger than trimers and contain preformed C-swapped portions, such as in pentamers and hexamers (see Figure 2F,G and [10,11]).

Hence, we might hypothesize a possible similar behavior for other pt-RNases known to dimerize or oligomerize through the swapping of their N-termini, such as bovine seminal (BS)-RNase [50,58], human RNase 1 variants [59,60,61,62], human angiogenin [63], or even amphibian ranpirnase (onconase) [64].

Finally, we propose that the relevance of the data found in this work may not be limited to RNases. It is known, indeed, that many proteins involved in amyloidogenic diseases, such as cystatin-C, prion, or β_2_-microglobulin, initiate self-association through the 3D-DS mechanism [65,66,67]. Moreover, the subsequent massive aggregation driving the so-called neurodegenerative “deposition diseases” is in some cases directly triggered by “runaway” or propagated 3D-DS as well [5,55,68]. Therefore, since fibrillation often requires a long lag phase, the possibility of interfering with the onset of the first oligomerization seed might help the potential amyloidogenic proteins to evade a fibrillation destiny. Considering the huge steps performed in characterizing molecules, small peptides, or small proteins able to interfere with mature fibrils [69], the possibility of blocking the initial 3D-DS event with protein engineering [70], peptides, or chemicals in general might in the future have important therapeutic rebounds.

## 4. Materials and Methods

### 4.1. Materials

Type XII-A RNase A, RNase S, β-mercapto-ethanol, sodium phosphate (NaPi), and thioflavin-T (ThT) were purchased from Sigma-Aldrich. Ethanol (EtOH) and acetic acid (HAc) were obtained from Merck-Millipore. Yeast RNA was obtained from Boehringer. RNase S-protein was produced by precipitating the S-peptide from RNase S with EtOH [41]. Other chemicals were used at the highest purity available.

### 4.2. RNase A Oligomerization and Storage of the Purified Oligomers

RNase A was induced to oligomerize upon lyophilization from 40% HAc, following the procedure described by Crestfield and colleagues [12]. The HAc-lyophilized sample was re-suspended in 0.05 M NaPi, pH 6.7, and then purified through cation-exchange Source 15S HR 16/10 or 10/10 column(s) attached to an ÄKTA-FPLC system (GE-Healthcare, Milan, Italy) [9,20]. After eluting the monomer (M) and the N-dimer (N_D_), respectively, with NaPi 0.07 M and 0.09 M, a linear 0.09–0.18 M NaPi concentration gradient was applied to elute the RNase A oligomers ranging from the C-dimer (C_D_) to N+C-swapped hexamers and large multimeric traces (Figure 1B) [20]. Each species was collected up to completely fill 50 mL “Falcon” tubes by adjusting the NaPi concentration to 0.10–0.12 M before storing the samples at 4 °C. In other experiments, the two dimers were separately collected while the oligomers eluted after the two dimers, i.e., from NC_T_ to large multimers, were collected together and stored at 4 °C as well until analysis, as described above.

### 4.3. Chromatographic Isolation of RNase A “Super Aggregates” (SAs)

After six to 15 months, the sample was concentrated with Amicon 3 ultrafilters (3 kDa cut-off, Merck-Millipore, Milan, Italy), and analyzed either with Superdex 75 or Superdex 200 HR 10/300 size-exclusion chromatography (SEC) columns attached to an ÄKTA-FPLC system (GE Healthcare). Columns were equilibrated with 0.20 M NaPi buffer at pH 6.7 [20], and mixtures eluted at a 0.10–0.12 mL/min flow rate. The resulting chromatographic patterns were analyzed with the Unicorn 5.01 Software (GE Healthcare), and each species (SAs or, better, SA-1,2,3) eluted from the dead volume columns was stored at 4 °C until new analyses were performed after concentrating it with the same Amicon ultrafilters. The SA concentration was spectrophotometrically estimated considering an ε^1%^_280_ = 7.3 [71], the same as that used either for RNase A monomer or small oligomers [9]. Considering the low yield of SAs produced and collected, we performed the totality of the analyses only with SA-2, i.e., the most abundant species, while the investigations of SA-1 and -3 were performed only when their amount made this possible.

### 4.4. Preliminary Characterization of SAs

The cuvette containing concentrated SAs was first analyzed visually to detect possible turbidity. Therefore, samples were centrifuged at 14,500 rpm for 8 min in a Mini-Eppendorf centrifuge (Eppendorf, Hamburg, Germany) before performing spectrophotometric measurements. Only the supernatant was spectrophotometrically analyzed while the pellet was stored for further studies. The A_260_/A_280_ ratio of the supernatant was calculated considering that its increase is proportional to the presence of large aggregates.

### 4.5. Estimation of SAs Enzymatic Activity

The enzymatic activity of SAs was estimated and compared with that of the RNase A monomer (M). We could not perform precise measurements because it was not possible to determine the correct amount of SAs introduced. The substrate was yeast RNA, a single-stranded (ss)-RNA. We followed the protocol described by Kunitz, for which the catalytic activity is proportional to the decrease in the yeast RNA Abs_300_ per min per µg of protein species [40]. Assays were performed with a Jasco V-650 spectrophotometer on 0.6 mg/mL (Abs_300_ ≈ 0.8) yeast ss-RNA solubilized in 0.10 M sodium acetate buffer, pH 5.0 [20]. The final concentration of each SA was only estimated in the reaction mixture to be between 15 and 20 µg/mL. The blank was Abs_300_ of the sodium acetate buffer while the negative control was Abs_300_ of the yeast RNA solution without the enzyme. The same analyses were performed in parallel with native monomeric RNase A (M, 0.8 µg/mL) for a qualitative comparison [9].

### 4.6. Non-Denaturing Cathodic PAGE

Non denaturing cathodic 12.5% acrylamide PAGE was performed following the protocol of Goldenberg [44], partially modified in [34], using ammonium persulfate and *N*,*N*,*N*′,*N*′-tetramethylethylenediamine (TEMED) despite riboflavin and UV-light. A Mini-Protean device (BioRad, Hercules, CA, USA), placed in an ice-bath and containing 0.35% β-alanine and HAc, pH = 4.0, was used [44]. The purified species were partially desalted with Amicon 3 ultrafilters, and each sample was analyzed without boiling and icing. The loading buffer was 0.05% methyl green (4x) dissolved in a 50% glycerol solution [44]. Gels were run for 90 min at 200 V, with the electrophoresis cell immersed in an ice-bath. Staining was performed with a 0.1% Coomassie Brilliant Blue and destaining with 10% HAc/20% EtOH solution.

### 4.7. Western Blot Analysis

To confirm whether SAs derived from RNase A, we tested their interactionusing the IgG anti-HP-RNase (IgTech, Salerno, Italy), which is known to also be active against RNase A [23]. The antibody, purified through immuno-affinity chromatography [72], was kindly provided by Prof. E. Pizzo (University of Naples, Federico II). About 5 to 15 μg of each RNase A M, N- and C-dimers, or estimated amounts of SA-2 were run with 12.5% SDS PAGE. Alternatively, non-denaturing cathodic PAGE was performed in the aforementioned Mini-Protean device, and then blotted for 60 min with a PVDF membrane (Immobilon-P) refrigerated in an ice-bath. The membrane was incubated with 5% BSA in a blocking TBST solution (50 mM TrisHCl, pH 7.5; 0.9% NaCl; 0.1% Tween20) for 1 h at room temperature (RT), and then probed overnight (ON) at 4 °C with anti-HP-RNase at a 1:1000 ratio. After TBST washing, the membrane was developed with the anti-rabbit IgG peroxidase-conjugate antibody (1:2000, Cell Signaling Tech, Danvers, MA, USA) and analyzed with a chemo-luminescent detection system (Immune-Star Western CKit, Bio-Rad, Hercules, CA, USA). Blotted proteins were then detected with a Chemi-Doc XRS Imaging System (Bio-Rad).

### 4.8. Thioflavin-T (ThT) Assays

Thioflavin-T (ThT) assays were carried out with SA-2 in parallel with RNase A monomer, dimers, trimers, large oligomers, and insoluble residues. ThT preferentially binds to amyloid fibrils, but, even though at less extent, also to pre-fibrillar or large oligomeric aggregates [32]. This induces a fluorescence quantum yield increase, and a blue-shift in λ_max_ [73]. Fibrils are characterized by a strong fluorescence emission increase at 482 nm, after 450 nm excitation, caused by the immobilization of the C-C bond connecting the ThT aniline and benzothiazole rings [74]. Notably, productive binding occurs for structures characterized by at least four consecutive β-strands [46].

The RNase A monomer (M) concentration was brought to about 50 µM (0.7 mg/mL) and 25 µM for the dimers, 15 µM for the trimers, and 10 µM for the tetramers; instead, the concentration of larger oligomers and SA-2 was estimated to be between 8 and 1 µM, or even lower. Samples were assayed with ThT according to the protocol described by LeVine [75]: 30 µL of each RNase A species was added to a 0.22-µm-filtered 25 µM ThT solution in 25 mM sodium phosphate buffer (NaPi), pH 6.0. Each sample was excited at 440 nm and the ThT fluorescence emission recorded in the 450–625 nm range. Measurements were performed in triplicate with a Jasco FP-750 Spectrofluorometer (Jasco Europe, Cremella (LC), Italy), thermostatically maintaining the cell holder at 25°C.

### 4.9. Investigations with Transmission Electron Microscopy (TEM)

Samples (10 μL) of RNase A multimers, SA-2, or insoluble residues were desalted with Amicon 3 ultrafilters, then separately dispersed on a 100-mesh copper grid coated with a Formvar film, and left to dry at RT. The grid was then stained for 1 min with 2% aqueous phosphotungstic acid (pH 6.8) negative staining. Samples were analyzed in a Philips Morgagni transmission electron microscope (Thermo-Fisher Sci., Waltham, MA, USA) operating at 80 kV and equipped with a Megaview III camera for digital image acquisition. The Radius software (EMSIS GmbH, Münster, Germany), implemented in the transmission electron microscope, was used for image acquisition. The morphometric evaluation of the precipitate size (diameter) was performed on randomly selected images for a total of 100 precipitates per sample. The mean value ± standard deviation (SD) was calculated for each sample. Precipitates were grouped in size classes, and the number of precipitates present in each class was plotted. Then, statistical analysis was performed by one-way Anova and the Bonferroni post-hoc test.

## 5. Conclusions

Under non-harsh conditions and after long-term storage, the N-termini-swapped-containing oligomers of RNase A, not the native monomer, formed large super-aggregates (SAs) that, at present, represent the RNase A species closest to pre-fibrillar ones. We hypothesized that RNase A controlled oligomerization [9,10,36,50], although this occurred under conditions far from super-saturation that in turn favor fibrillation [25], can be considered a preliminary step towards an evolution “wishing” an amyloid destiny. However, we observed that contemporary regression to monomer and the “self-chaperone” nature of RNase A [26] strongly limits the extent of SAs formation as well.

Nevertheless, the data collected suggest that the opening of the “per se” amyloidogenic RNase A N-terminal domain, in turn allowing the formation of N-swapped oligomers, might facilitate a potentially massive aggregation of the enzyme. Therefore, the possibility to delay the initial 3D-DS self-association event might open valuable therapeutic strategies in order to contrast pathologies induced by the 3D-DS affecting amyloidogenic proteins.

## Figures and Tables

**Figure 3 ijms-23-11192-f003:**
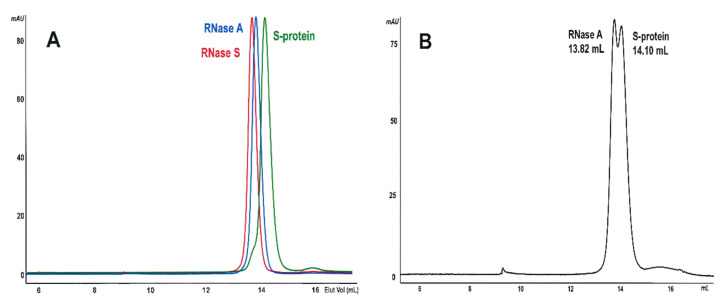
SEC analysis of native RNase A and its subtilisin-protease derivatives. RNase A species were injected onto a Superdex 75 HR 10/300 column equilibrated with 0.2 M NaPi buffer, pH 6.7. (**A**) Overlap of the three patterns obtained with 0.25 mg of RNase A (blue curve), RNase S (red) or S-protein (green), with a flow rate of 0.20 mL/min. (**B**) Analysis of 0.2 + 0.2 mg RNase A + S-protein mixture, with a flow rate of 0.10 mL/min.

**Figure 5 ijms-23-11192-f005:**
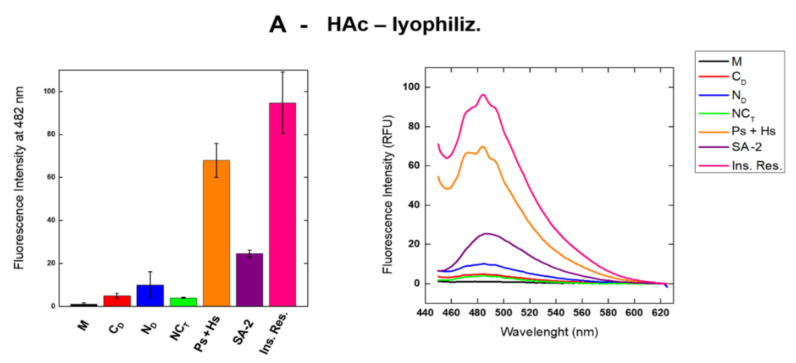
Fluorescence emission intensity of RNase A and its supramolecular derivatives upon interaction with Thioflavin-T (Th-T). Fluorescence was measured at 482 nm after excitation at 450 nm. (**A**) RNase A species recovered after 40% HAc-lyophilization, or (**B**) 60 °C heating of 40% EtOH solutions of the enzyme; (**Right**) panels: fluorescence intensity of RNase A monomer, dimers, trimers, pentamers+hexamers, SA-1+2 mix, and the insoluble residues (Ins. Res.) withdrawn from SAs. (**Left**) panels: histograms reporting the relative maximum value of each intensity curve measured at 482 nm reported in the right panels. Curves and relative maxima ± S.D. in the histograms are the average of three repeated tests.

**Figure 6 ijms-23-11192-f006:**
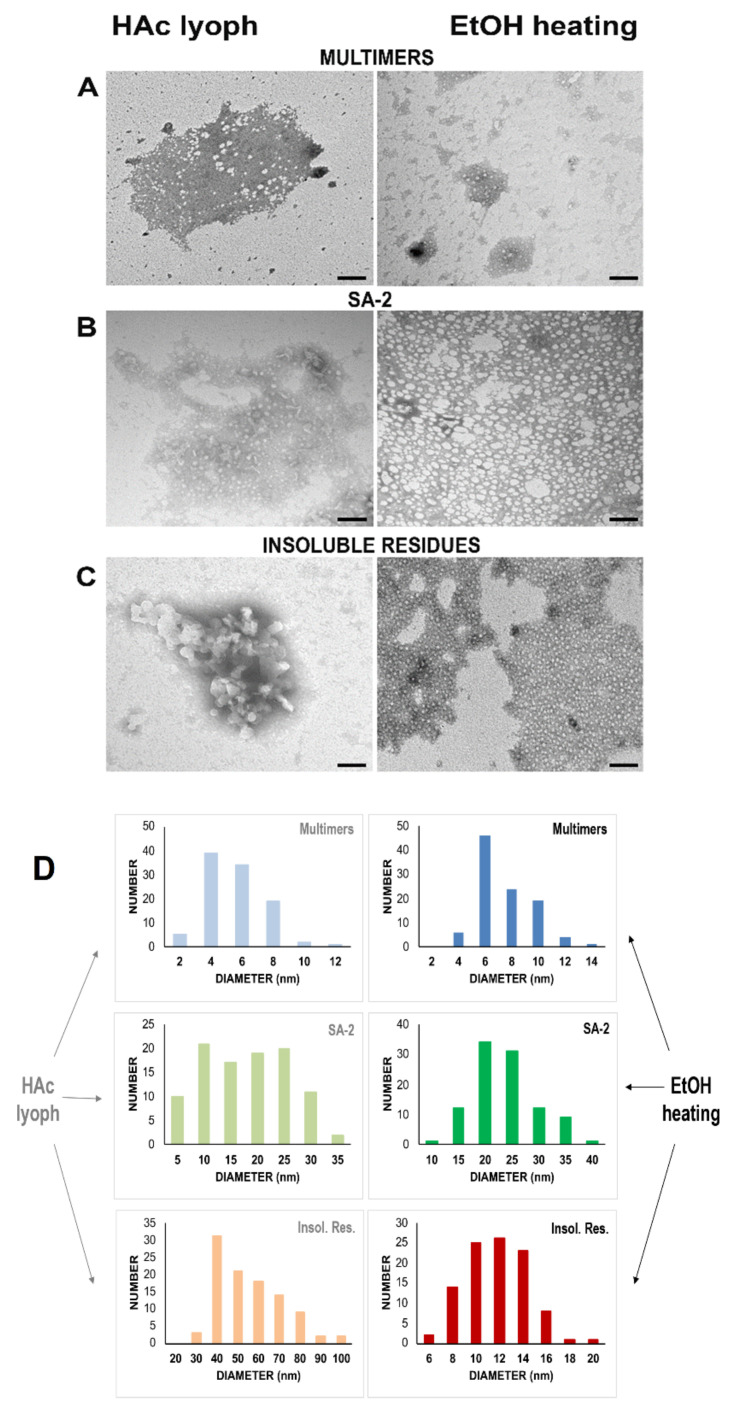
TEM analysis of RNase A multimers, SA-2, and insoluble residues. Representative images of transmission electronic microscopy (TEM) of RNase A multimers (**A**), SA-2 (**B**), and insoluble residues (**C**), obtained after applying both HAc lyophilization or EtOH thermal incubation conditions. Note the circular species in all samples. Bars, 100 nm. (**D**) Diameters’ size distribution of the RNase A species analyzed in panels (**A**–**C**): the diameters are grouped in size classes of 2 (multimers in both conditions), 5 (SA-2 in both conditions), and 10 nm (insoluble residues in HAc lyophilization), or 2 nm (insoluble residues in EtOH thermal incubation). The number of circular species detected in each class is plotted.

## Data Availability

Not applicable.

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
