# Peer review of "Slow Evolution toward “Super-Aggregation” of the Oligomers Formed through the Swapping of RNase A N-Termini: A Wish for Amyloidosis?"

_ijms, 2022, doi:10.3390/ijms231911192_

Round 1
Reviewer 1 Report
The authors reported an interesting discovery that RNase A can evolve into an amorphous like non-amyloid aggregate through domain swap mechanism after more than 1 year incubation. They used a variety of biophysical techniques to support their argument. In general, this paper can be published in IJMS. Yet, some improvements need to be done before its final publication.
(1) Please provide some details regarding to the story behind the discovery. Was the discovery found by accidently or on purpose? The incubation time is so long.
(2) By what standard, the authors identify the aggregates with the same MW (e.g. 27372 in table 1) to be either N-dimer (ND) or C-dimer (CD).
(3) Using what program, the 3D structure of Rnase A was generated in Fig. 7B. Are they real structure or just for illustration purpose?
(4) The tentative model in Fig. 7 B is difficult to understand. Please provide more details and more background knowledge to describe this model.
(5) Using what techniques, the N-C swap and C-swap are differentiated?
(6) Please provide some discussions about the implication of this discovery for the prevention of amyloid diseases.
Author Response
Please, see attachment.

Reviewer 2 Report
I believe the manuscript in its current format is nicely written and will be of importance to all researchers interested in amyloidosis.
My only recommendation is to improve the quality of Figures 2 and 6.
Author Response
Please, see the attachment

Reviewer 3 Report
The following questions are listed here to be addressed:
1. Can the author list the PDB codes of the molecular structures for the RNase-A oligomers they used in caption of Fig. 2?
Minor comments:
Labels in Figs. 1L and 1L are hard to see.
The Y axis label of Fig. 6D are hard to see.
Author Response
Please see th attachment

Reviewer 4 Report
Gotte et al. reported an interesting observation that RNase A oligomerizes after prolonged incubation or at high temperature and in high concentration. They performed a series of biophysical studies to support their arguments and proposed a potential structural model of the aggregates. The narrative is well told and the English usage is professional. However, I would suggest the authors pay attention to the following comments:
(1) After reading the manuscript, I am still a bit lost on why the readers should pay attention to the RNase A aggregation. I would suggest the authors include potential applications of the observations (oligomerization/aggregation of this enzyme) to therapeutic development or other fields
(2) Since the concentrations of larger oligomers and SA-2 used in the ThT assay are not accurately measured, how can one make sure the oligomer species used in the ThT assay (Figure 5) are quantitatively similar? If not, the low ThT intensity itself may not be able to rule out the possibility of fibril formation.
(3) It is recommended to highlight (or color-code) the peaks that correspond to the oligomers of interest (L.M., Hs, Ps, etc.) in the chromatograms.
(4) Explain in Figure 7 caption why the boxes indicating stabilizing interactions are colored differently.
(5) It is better to include ladder lanes in SDS-PAGE and Western Blots.
